# Easy access to medium-sized lactones through metal carbene migratory insertion enabled 1,4-palladium shift

Yinghua Yu[1,5], Pushkin Chakraborty[1,5], Jinshuai Song [2,3,5], Lei Zhu[1], Chunsen Li[2]* & Xueliang Huang [1,4]*

Reactions that efficiently construct medium-sized lactones are significant, as they overcome the unfavorable entropic factor and transannular interactions for ring closure, and the lactones produced are common structural motifs recurring in many biologically active compounds. Herein, we describe a valuable strategy for medium-sized lactone synthesis by accomplishing site-selective C–H bond functionalization via a palladium carbene migratory insertion enabled 1,4-palladium shift. The overall process achieves the formal dimerization of two readily available benzaldehyde derivatives, providing value-added products medium-sized lactones. Our method is amenable to late-stage modification of approved drugs and other complex molecules. Mechanistic studies including deuterium-labeling experiments and DFT calculation shed light on the reaction pathways.

[1] Key Laboratory of Coal to Ethylene Glycol and Its Related Technology, Center for Excellence in Molecular Synthesis, Fujian Institute of Research on the Structure of Matter, Chinese Academy of Sciences, 350002 Fuzhou, Fujian, China. [2] State Key Laboratory of Structural Chemistry, Fujian Institute of Research on the Structure of Matter, Chinese Academy of Sciences, 350002 Fuzhou, Fujian, China. [3] College of Chemistry, and Institute of Green Catalysis, Zhengzhou University, 450001 Zhengzhou, China. [4] State Key Laboratory of Elemento-Organic Chemistry, Nankai University, 300071 Tianjin, China. [5]These authors contributed equally: Yinghua Yu, Pushkin Chakraborty, Jinshuai Song. *email: chunsen.li@fjirsm.ac.cn; huangxl@fjirsm.ac.cn

Palladium/hydride shift[1–3] is an attractive strategy to achieve selective inert C–H bond activation[4–12], while without the requirement of installing a directing group (Fig. 1a). This migration event provides a convenient way to activate a specific C–H bond, which might be difficult to achieve by conventional methods. However, for such a fundamental transformation, most studies are focused on migration proceeded within a single molecule[13–27], which means the reactants involved have to be carefully designed to possess a fitted geometry to undergo the desired migration. Elegant exceptions were reported by Larock and coworkers almost 20 years ago (Fig. 1b)[28–30], in which the migration of the palladium moiety took place after an intermolecular reaction of two readily available reactants. The insertion of alkyne was crucial to construct the vinylpalladium intermediate that was ready to undergo corresponding 1,4-palladium migration.

Many feedstock reagents do not contain any directing groups. For related reactions of these compounds, the palladium metal center is too close to the reacting site. The unfavored strain energy makes it is inaccessible to form a palladacycle via direct C–H bond palladation[31,32]. Inspired by the seminal work from Larock[28–30], we propose that whether we can use a modular one-carbon synthon to facilitate the formation of a thermodynamically stable five-membered palladacycle, which in turn realizes the selective C–H bond activation (Fig. 1c). We believe that the bridging of such a unit to simple reactants may open an avenue for functionalization of inert C–H bonds.

In line with our research interests in diazo compounds involved C–H bond activation[33–35], we assumed that a palladium carbene-enabled acylation[34] for dibenzo-fused seven-membered lactone synthesis could be an ideal probe to test our assumption (Fig. 1d). In this designed reaction, both reactants could be prepared in multi-gram scale from readily available salicylaldehyde analogs. Studies on palladium carbene participated cross-coupling reactions[36–40] support the feasibility of our hypothesis. According to recent discovery, tricyclic ring systems possessing a dibenzo structure joined to a central seven-membered lactone show the potential for preventing or treating malignant diseases[41]. Eurotinone, an effective KDR kinase inhibitor, possesses the exact tricyclic backbone. Moreover, modular synthesis of these lactones based on traditional methods, such as Baeyer-Villiger (BV) oxidation[42] and macrolactonization[43], is challenging. As for the BV reaction, the regioselectivity is low when unsymmetric cyclic

ketones are employed, and for the latter, multi-step manipulation is required for the synthesis of the particular reactants. As a common feature of intramolecular reactions, the diversity of the products is limited by the ease of synthesis of the reactants. Thus, it is hard to offer large number of samples that are required for massive screening and evaluation by these traditional methods. Here, we report a palladium carbene migratory insertion-enabled medium-sized lactone synthesis[44]. In this reaction, diazo compounds generated in situ are employed as modular dockable building blocks to promote the C–H bond activation. Moreover, this methodology was found to be efficient for late-stage functionalization of complex molecules, which could be useful for fragment-based drug discovery[45].

## Results

**Reaction development**. To validate our hypothesis, $N$-tosylhydrazone **2a** was selected as the precursor of bifunctional diazo compound to react with 2-bromobenzaldehyde **1a**. When the reaction was carried out in THF (tetrahydrofuran) at 80 °C for 10 h, using Pd$_2$(dba)$_3$•CHCl$_3$ (2.5 mol%) as palladium source, dppm (bis(diphenylphosphanyl)methane, 7.5 mol%) as ligand, K$_3$PO$_4$ as base, the expected seven-membered lactone **3** was indeed formed in 5% nuclear magnetic resonance (NMR) yield (Table 1, entry 1). Initial examination of different bidentated phosphine ligands (Table 1, entries 1–10) found that Xantphos ((9,9-dimethyl-9H-xanthene-4,5-diyl)bis(diphenylphosphane)) performed the best, affording **3** with 80% NMR yield (Table 1, entry 10). After a brief survey of other palladium sources (Table 1, entries 11–13) (for a detailed optimization study, see Supplementary Information Table S1 to Table S4), we found Pd(OAc)$_2$ (5 mol%) was ideal. The reaction could complete in a relatively short time, and the corresponding lactone **3** was obtained in 76% isolated yield (Table 1, entry 11). $o$-Iodobenzaldehyde **1b** could participated in the palladium-catalyzed lactonization as well, the seven-membered lactone **3** was produced in slightly lower NMR yield (entry 14).

**Substrate scope**. With the optimized reaction conditions in hand (Table 1, entry 11), we sought to explore the substrate scope with respect to different $o$-bromobenzaldehydes and $N$-tosylhydrazones bearing hydroxy tether (Table 2). As depicted, a variety of substituted $o$-bromoarylaldehydes **1** could participate in current lactonization reaction regardless of the electronic nature of

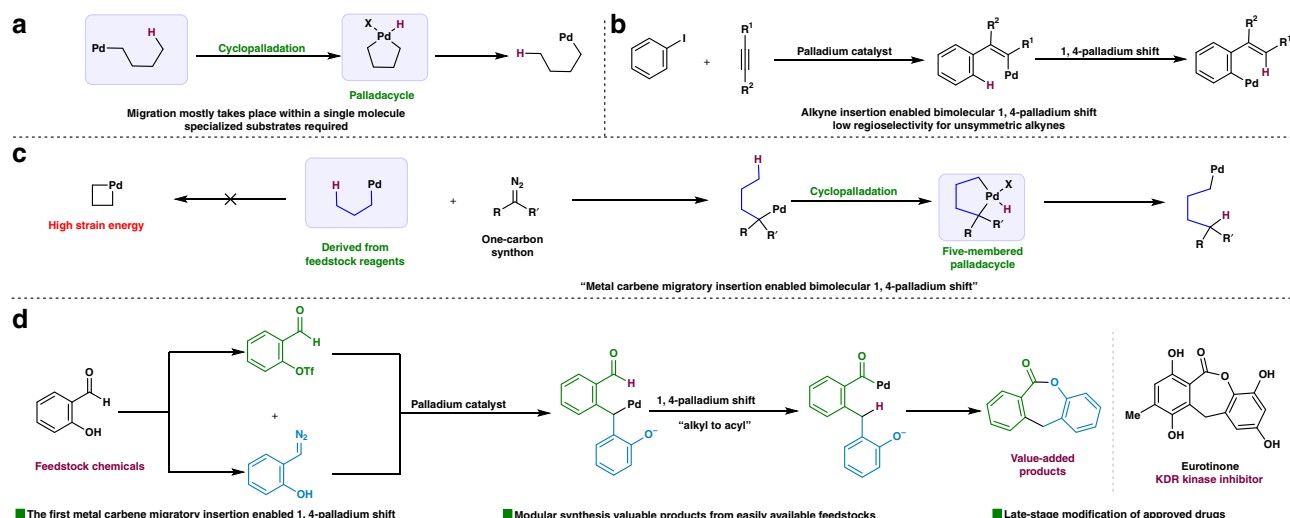

**Fig. 1 Reaction modes for palladium-catalyzed C–H bond functionalization. a** Reaction mode for 1,4-palladium shift. **b** Alkyne insertion-enabled 1,4-palladium shift. **c** Proposed metal carbene migratory insertion-enabled 1,4-palladium shift. **d** Working hypothesis for seven-membered lactone synthesis.

**Table 1 Optimization of reaction conditions for seven-membered lactone synthesis.**

| Entry[a] | [Pd], x mol% | Ligand, 7.5 mol% | Base, z equiv. | Yield/%[b] |
|---|---|---|---|---|
| 1 | Pd$_2$(dba)$_3$•CHCl$_3$, 2.5 | dppm | K$_3$PO$_4$, 4.0 | 5 |
| 2 | Pd$_2$(dba)$_3$•CHCl$_3$, 2.5 | dppe | K$_3$PO$_4$, 4.0 | 52 |
| 3 | Pd$_2$(dba)$_3$•CHCl$_3$, 2.5 | dppp | K$_3$PO$_4$, 4.0 | 67 |
| 4 | Pd$_2$(dba)$_3$•CHCl$_3$, 2.5 | dppb | K$_3$PO$_4$, 4.0 | 57 |
| 5 | Pd$_2$(dba)$_3$•CHCl$_3$, 2.5 | dpppe | K$_3$PO$_4$, 4.0 | 35 |
| 6 | Pd$_2$(dba)$_3$•CHCl$_3$, 2.5 | dppf | K$_3$PO$_4$, 4.0 | 49 |
| 7 | Pd$_2$(dba)$_3$•CHCl$_3$, 2.5 | DPEphos | K$_3$PO$_4$, 4.0 | 5 |
| 8 | Pd$_2$(dba)$_3$•CHCl$_3$, 2.5 | Xantphos | K$_3$PO$_4$, 4.0 | 77(76)[c] |
| 9 | Pd$_2$(dba)$_3$•CHCl$_3$, 2.5 | Xantphos | K$_3$PO$_4$, 4.0 | 77[d] |
| 10 | Pd$_2$(dba)$_3$•CHCl$_3$, 2.5 | Xantphos | K$_2$CO$_3$, 4.0 | 70(74)[c] |
| 11 | Pd(OAc)$_2$, 5 | Xantphos | K$_2$CO$_3$, 3.0 | 76(76)[c,e] |
| 12 | PdCl$_2$, 5 | Xantphos | K$_2$CO$_3$, 3.0 | 73[e] |
| 13 | [(η-C$_3$H$_6$)PdCl]$_2$, 2.5 | Xantphos | K$_2$CO$_3$, 3.0 | 75[e] |
| 14 | Pd(OAc)$_2$, 5 | Xantphos | K$_2$CO$_3$, 3.0 | 64[f] |

[a]Reaction condition: **1a** (0.2 mmol), **2a** (0.4 mmol), [Pd] (5 mol%), Ligand (7.5 mol%), base (z equiv.) in THF (2.0 mL), stirring under atmosphere of Argon at 80 °C for 24 h
[b]NMR yields were determined using mesitylene as internal standard
[c]Isolated yield
[d]Dioxane
[e]12 h
[f]**1b** was employed instead of **1a**

substituents incorporated on the phenyl ring. The corresponding lactones were obtained in moderate to high yields (products **3–13**, 65%–84% yields). Sterically hindered 1-bromo-2-naphthaldehyde was also a viable substrate for the reaction, leading to the corresponding product **14** in 55% isolated yield.

Additional experiments revealed that the current protocol for seven-membered lactone synthesis was efficient, as a wide array of N-tosylhydrazones derived from salicylaldehyde analogs could also react well with 2-bromobenzaldehyde **1a** (products **15–37**). Electron-donating (methyl and methoxy, products **15**, **20**, **21**, and **25**) and electron-withdrawing (chloro, fluoro, even bromo, products **16–19**, **22–24**, and **26–27**) groups decorated on the aryl ring were tolerated, regardless of the position and steric effects. The tolerance of chloro and bromo groups has offered convenient handle for further transition-metal catalyzed cross-coupling reactions. Functional moieties, such as carbon–carbon triple bond and labile silyl group, stayed intact, and corresponding lactones **28** and **29** were isolated in 67% and 81% yields, respectively. It is worthwhile to mention that N-tosylhydrazone derived from corresponding hydroxylketone was also a viable bifunctional carbene precursor, giving the substituted lactone in moderated yield under a slightly modified condition (product **30**). The establishment of a chiral carbon center shows the synthetic potential of current method in catalytic asymmetric synthesis, which could be our future research objective. The current lactonization shows potential application on the area of

fragment-based drug discovery. As o-hydoxy-N-tosylhydrazones derived from bio-relevant molecules, such as methylparaben, paracetamol, carvacrol, thymol, eugenol, estrone and methyl N-Phth-L-tyrosinate, were competent diazo precursors, and the potentially bioactive ε-lactones **31–37** were obtained in moderate to good yields.

To further demonstrate the synthetic potential, we proceeded to examine the applicability of our designed chemistry for o-pseudo-halo benzaldehyde substrates instead of 2-bromobenzaldehyde. In principle, both reactants could be availed by exploiting 2-hydroxy benzaldehyde as the same starting material. Treatment of 2-formylphenyl trifluoromethanesulfonate **1c** and **2** under the previous optimized conditions (Table 1, entry 11) produced **3** with 53% of GC yield. Either changing Xantphos to other biphosphine ligands or using other bases instead of K$_2$CO$_3$ did not improve the results. The best result (80% of isolated yield) was obtained when Pd(OAc)$_2$ was replaced by [Pd(η-C$_3$H$_6$)Cl]$_2$ (2.5 mol%) (for a detailed condition experiments, see Supplementary Information Tables S5 to Table S8).

A feature for current transformation by using o-pseudo-halo aryl aldehydes as the electrophiles is the implementation of formal dimerization of abundant salicylaldehyde analogs to a range of functionalized seven-membered lactones. Under the optimized conditions we found that both reactants derived from 2-hydroxy benzaldehydes containing a variety of substituents on the phenyl ring could be formally dimerized, giving

**Table 2 Substrate scope with respect to *o*-bromoarylaldehydes 1 and *N*-tosylhydrazones 2ᵃ.**

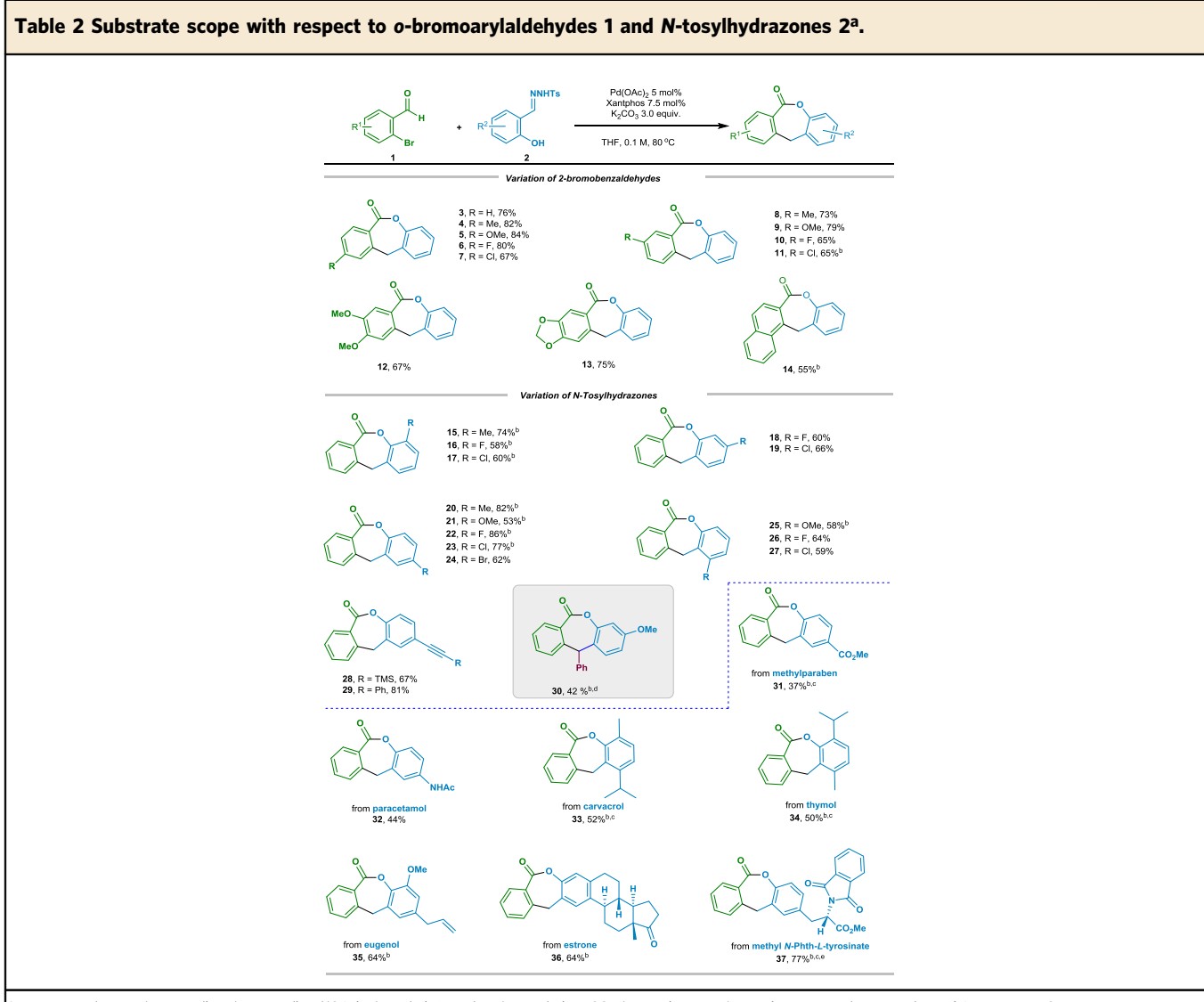

ᵃReaction condition: **1** (0.2 mmol), **2** (0.4 mmol), Pd(OAc)₂ (5 mol%), Xantphos (7.5 mol%), K₂CO₃ (3 equiv) in THF (2.0 mL), stirring under atmosphere of Argon at 80 °C
ᵇPd(OAc)₂ 10 mol%, Xantphos 15 mol%
ᶜ90 °C
ᵈ100 °C
ᵉThe reaction was carried out in dioxane

corresponding lactones in moderate to high yields (Table 3, products **3**, and **38**–**42**). Notable examples are those reactants possessing potentially reactive bromo group, could participate in current formal dimerization smoothly (product **41**). Similarly, the formal cross-dimerization of salicylaldehyde analogs also exhibits very general substrate scope, diverse substituents including electron-donating and electron-withdrawing groups were tolerated, and displayed good orthogonal reactivities of the triflates and *N*-tosylhydrazones (Table 3, products **5**, **9** and **43** to **55**). Alkynyl group and labile trimethylsilyl (TMS) moiety stayed intact under standard reaction conditions (products **56** and **28**).

Current strategy was found applicable for more challenging eight-membered lactone synthesis. Under the optimal conditions for the seven-membered lactone synthesis, we indeed observed the formation of desired lactone **58** (*n* = 1) from the reaction mixture, albeit in 15% NMR yield. After identification of the effects of phosphine ligands, palladium sources and bases (for a detailed optimization study, see Supplementary Information Tables S9–S13), the best result with 76% of isolated yield of **58** was obtained with dppb (1,4-bis(diphenylphosphanyl)butane) at 5 mol% Pd loading. It is worthwhile to mention that, for this

specific reaction, we have observed the formation of dihydroisobenzofuran as side-product in small amount (5% isolated yield under optimized conditions). This outcome is noteworthy, as the formation of eight-membered lactone **58**, which is generally considered to be energetically less favorable, could override the seemingly favorable pathway for the formation of five-membered compound. This unusual chemo selectivity turned out to be quite general under current conditions for a range of eight-membered lactone synthesis. As the examples shown in Table 3 (products from **57** to **63**), lactone formation was not significantly affected by the presence of electron-donating or -withdrawing groups on the phenyl ring of the triflates. As exemplified by the products with strongly electron-withdrawing group (product **59**) and with weakly electron-withdrawing group (product **60**) were isolated in 70% and 51% yields, respectively. The reactions for the substrates bearing weakly electron-donating group (products **61** and **63**) and with strongly electron-donating group (product **62**) proceeded well. Similarly, functional groups, such as TMS and alkynyl groups (product **57**) were compatible.

The abundance of phenolic-derivatives and well-established procedure of *ortho*-formylation of phenols[46] offered us the

**Table 3 Substrate scope for formal dimerization of salicyaldehyde analogs.[a]**

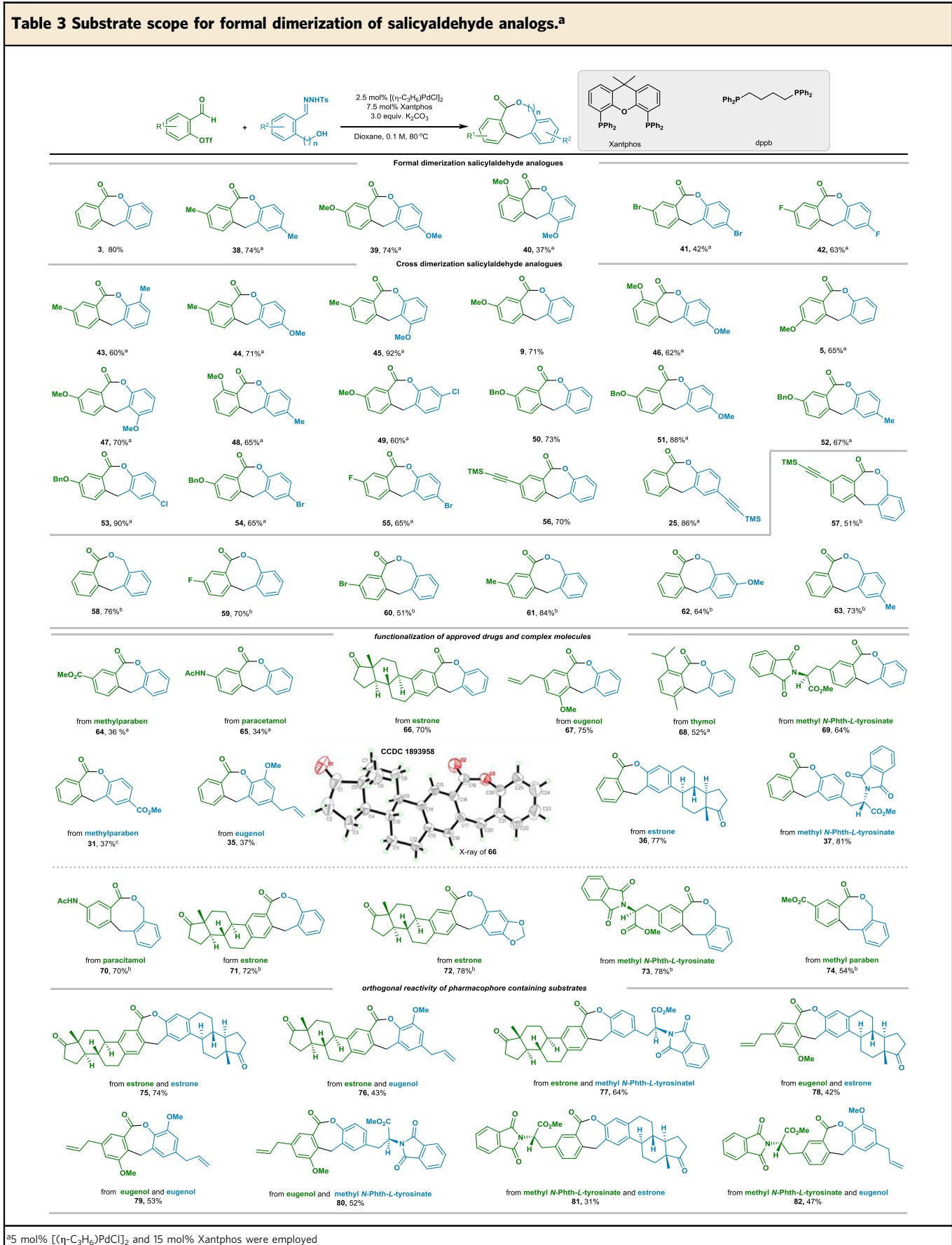

Formal dimerization salicylaldehyde analogues

**3**, 80%  **38**, 74%[a]  **39**, 74%[a]  **40**, 37%[a]  **41**, 42%[a]  **42**, 63%[a]

Cross dimerization salicylaldehyde analogues

**43**, 60%[a]  **44**, 71%[a]  **45**, 92%[a]  **9**, 71%  **46**, 62%[a]  **5**, 65%[a]

**47**, 70%[a]  **48**, 65%[a]  **49**, 60%[a]  **50**, 73%  **51**, 88%[a]  **52**, 67%[a]

**53**, 90%[a]  **54**, 65%[a]  **55**, 65%[a]  **56**, 70%  **25**, 86%[a]  **57**, 51%[b]

**58**, 76%[b]  **59**, 70%[b]  **60**, 51%[b]  **61**, 84%[b]  **62**, 64%[b]  **63**, 73%[b]

functionalization of approved drugs and complex molecules

from **methylparaben** **64**, 36 %[a]

from **paracetamol** **65**, 34%[a]

from **estrone** **66**, 70%

from **eugenol** **67**, 75%

from **thymol** **68**, 52%[a]

from **methyl N-Phth-L-tyrosinate** **69**, 64%

from **methylparaben** **31**, 37%[c]

from **eugenol** **35**, 37%

CCDC 1893958

X-ray of **66**

from **estrone** **36**, 77%

from **methyl N-Phth-L-tyrosinate** **37**, 81%

from **paracitamol** **70**, 70%[b]

form **estrone** **71**, 72%[b]

from **estrone** **72**, 78%[b]

from **methyl N-Phth-L-tyrosinate** **73**, 78%[b]

from **methyl paraben** **74**, 54%[b]

orthogonal reactivity of pharmacophore containing substrates

from **estrone** and **estrone** **75**, 74%

from **estrone** and **eugenol** **76**, 43%

from **estrone** and **methyl N-Phth-L-tyrosinatel** **77**, 64%

from **eugenol** and **estrone** **78**, 42%

from **eugenol** and **eugenol** **79**, 53%

from **eugenol** and **methyl N-Phth-L-tyrosinate** **80**, 52%

from **methyl N-Phth-L-tyrosinate** and **estrone** **81**, 31%

from **methyl N-Phth-L-tyrosinate** and **eugenol** **82**, 47%

[a]5 mol% [(η-C$_3$H$_6$)PdCl]$_2$ and 15 mol% Xantphos were employed
[b]5 mol% [(η-C$_3$H$_6$)PdCl]$_2$ and 7.5 mol% dppb were employed
[c]The reaction was carried out at 100 °C

opportunity to enrich the substrate scope with lactones having different functionalities. As depicted, the triflates bearing various pharmacophore fragments, such as methylparaben, paracitamol, estrone, eugenol, thymol and methyl *N*-Phth-*L*-tyrosinate, could react with salicylaldehyde-derived *N*-tosylhydrazone under standard conditions, giving the corresponding functionalized bio-relevant seven-membered lactones in 34% to 75% isolated yields (products from **64** to **69**). For comparison, *N*-tosylhydrazones derived from methylparaben, eugenol estrone and *N*-Phth-*L*-tyrosinate were also tested as the precursors of diazo compounds to react with triflate **1c**. Again, the corresponding lactones could be prepared efficiently (Table 3, products from **31** to **37**, up to 81% isolated yield after column chromatography on silica gel). These potentially bio-relevant reactants were competent substrates for eight-membered homo-analogs synthesis as well. The corresponding lactones containing paracetamol (product **70**), estrone (products **71** and **72**), *N*-Phth-*L*-tyrosine (product **73**) and methylparaben (product **74**) motifs were prepared in moderate to high yields (54–78%). Unambiguous proof of structure and absolute configuration of the bio-relevant lactone **66** was achieved by single-crystal X-ray analysis.

Notably, this fragment-based technology was found to be applicable to couple two pharmacophore fragments, giving a variety of complex molecules in a modular fashion (Table 3, products from **75–82**). The proof-of-concept was firstly expressed by the formal dimerization of estrone and eugenol derivatives under the standard conditions, giving corresponding dimers (**75** and **79**) with seven-membered lactone linkers in 74% and 53% isolated yields, respectively. Subsequently, we examined the orthogonal reactivity of both reactants containing pharmacophore structural motifs. Triflate derived from estrone could react with the corresponding *N*-tosylhydrones prepared from eugenol and methyl *N*-Phth-*L*-tyrosinate, and the cross dimerized lactones **76** and **77** were obtained in 43% and 64% isolated yields after chromatography. Following the similar reaction design, highly complex molecules **78**, **80**, **81**, and **82** containing seven-membered lactone scaffolds could be obtained conveniently.

**Mechanistic investigation**. In order to gain insights about the reaction mechanism, representative isotopic labeling experiments

were carried out. Under otherwise identical conditions, triflate **1c** reacted with 2-hydroxy *N*-tosylhydrazone **2a** in presence of 10 equiv. of $D_2O$, **3** without any incorporation of deuterium atom was isolated in 55% yield (Fig. 2a). Whereas, when compound **[D]-1c** treated with hydrazone **2a** under the standard condition, **[D]-3** was produced (Fig. 2b). Finally, the relatively lower value of $k_H/k_D$ indicated C–H bond cleavage in the aldehyde moiety was not involved in the rate-determining step[47] (Fig. 2c).

Taken all the data together, we have proposed a reasonable mechanism for current medium-sized lactone synthesis (Fig. 2d). The reaction is initiated by oxidative addition of palladium(0) catalyst to *o*-pseudo-halo benzaldehyde to generate palladium(II) intermediate **I**. **I** could be considered as a receptor to react with bifunctional diazo compound, giving palladium-carbene intermediate **II**. Migratory insertion of **II** would produce intermediate **III**, which possesses the exact geometry that is ready to undergo a 1,4-palladium/hydride shift, thus achieving a selective C–H bond activation of the aldehyde moiety[48]. We anticipate such a reaction mode may not be limited to C–H bond activation of an aldehyde moiety. Other functional moieties possessing similar geometry could be applicable for related metal migration. The step for 1,4-palladium migration proceeds in an irreversible manner, since we did not observe any deuterium atom scrambling during the isotopic labeling experiments (Fig. 2b, c). The fact that no deuterium labeling of **3** occurred when the reaction was carried out in presence of 10 equiv. $D_2O$ (Fig. 2a) indicates that H-D exchange between of the potential palladium intermediates and the reaction media is relatively slow. Ring closure of **IV** generates a high-value medium-sized lactone, where two readily available aldehyde derivatives have been formally dimerized. In the case of eight-membered lactone synthesis, reductive elimination of intermediate **III** (*n* = 1) would give the side-product dihydroisobenzofuran.

**Density functional theory (DFT) calculations**. To get more details on the mechanism, a theoretical study of the energy surface of the proposed pathway for the reaction of **1a** and **2a** catalyzed by a palladium catalyst generated from $Pd(OAc)_2$ and dppb was carried out by DFT calculations. As shown in Fig. 3, a weak interaction between Pd complex and reactant **1a** is virtually thermoneutral ($\Delta G = 2.4$ kcal/mol). Oxidative addition of **1a** to

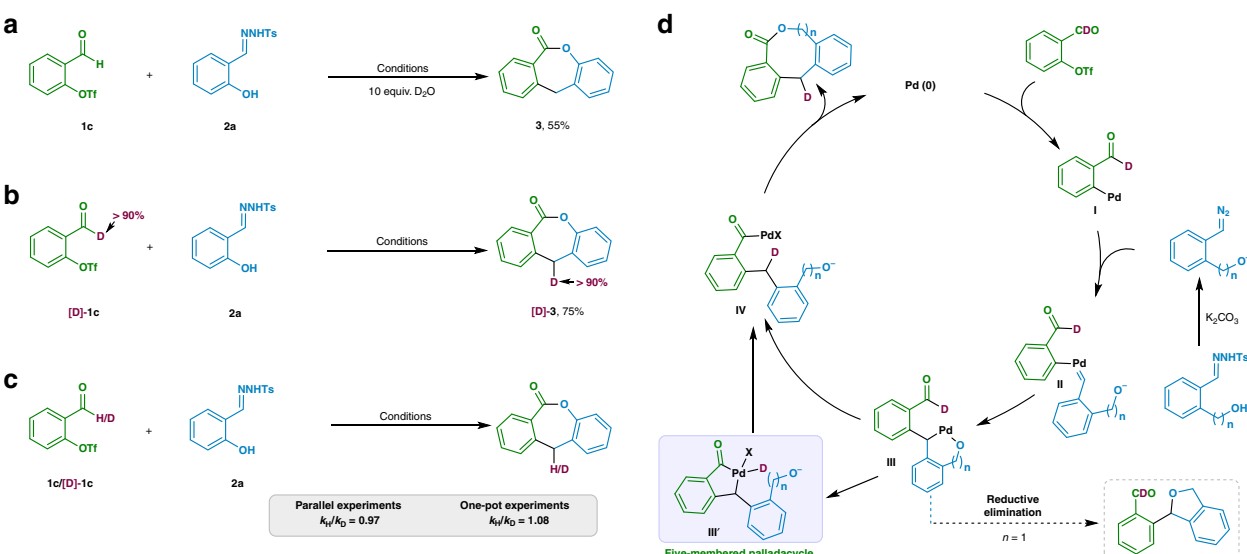

**Fig. 2 Mechanistic experiments. a** Reaction proceeded in presence of 10 equiv. of $D_2O$. **b** Deuterium-labeling experiment as a probe for 1,4-palladium migration. **c** Kinetic isotopic effect experiment for C–H bond cleavage in aldehyde **1c**. **d** Proposed catalytic cycle for palladium-catalyzed medium-sized lactone synthesis. X stands for possible counter anion.

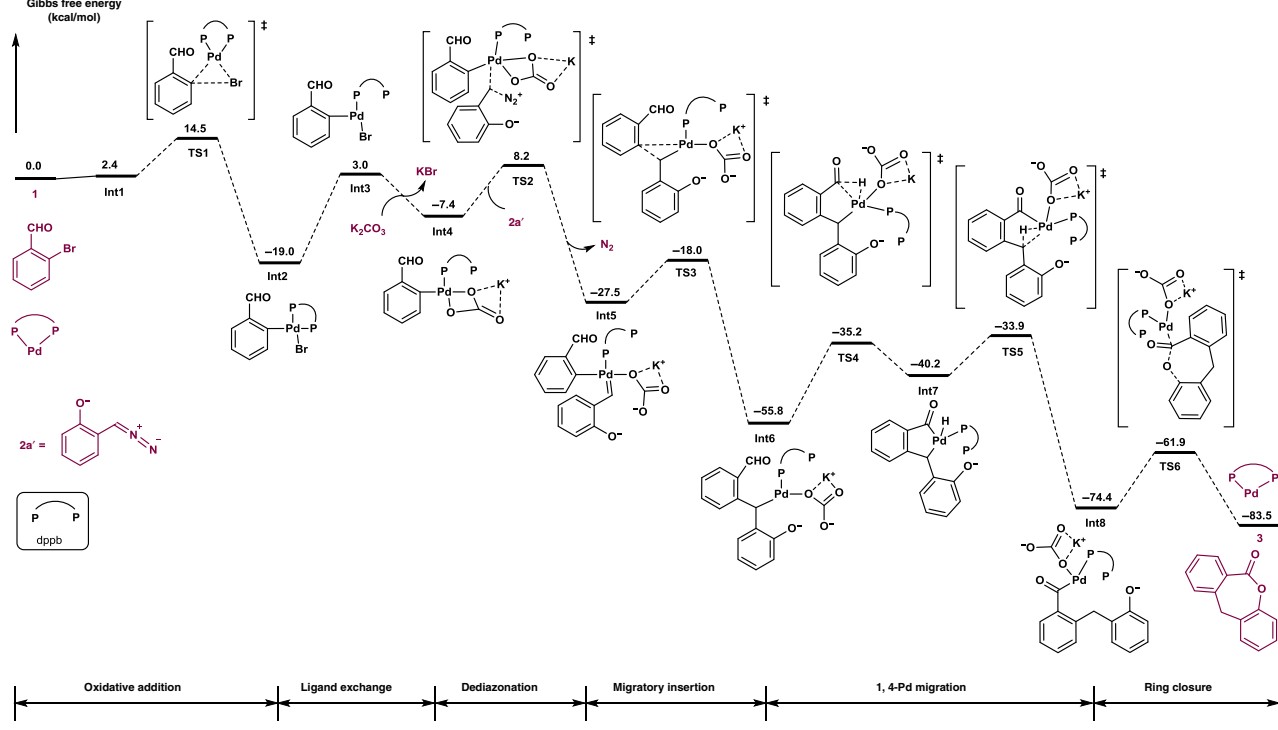

**Fig. 3 DFT calculations.** Reaction energy profiles calculated at M06/def2-TZVP//B3LYP/6-31G(d)(LANL2DZ) level.

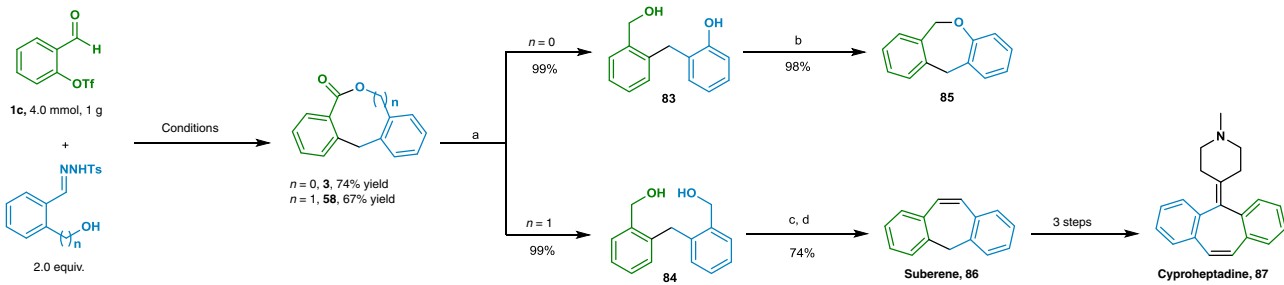

**Fig. 4 Synthetic manipulations.** Reagents and conditions: **a** LiAlH₄, THF, 0 °C, 3 h; **b** Ph₃P, diethyl diazene-1,2-dicarboxylate, toluene, 70 °C, 6 h; **c** DMP, ᵗBuOH, DCM, RT, 5 h; **d** TiCl₄, Zn, pyridine, THF, 0–80 °C, 6 h.

form **Int2** is thermodynamically downhill by 21.4 kcal/mol and requires an energy barrier of 12.1 kcal/mol (**TS1**). To provide a vacant site for further reaction, dissociation of one site of the bidentated ligand is endothermic of 22.0 kcal/mol, yielding **Int3**. Ligand exchange of bromide by K₂CO₃ to form **Int4** is an exothermic process. According to our study, the dissociation of dinitrogen is not that facile. Compared to **Int2**, the reaction of **2a'** with **Int4** has to overcome a barrier of 27.2 kcal/mol (**TS2**) in energy to generate palladium carbene species **Int5**. A subsequent migratory insertion takes place, giving **Int6** via a three-membered transition state **TS3**, and encountering a low barrier of 9.5 kcal/mol. The C–H activation process starts from **Int6**. Palladium atom inserts into the C–H bond of the aldehyde moiety, and produces **Int7**, which possesses a five-membered palladacycle with a hydride species sitting on the palladium center. This step presents a barrier of 20.6 kcal/mol (**TS4**). Hydride transfer from the palladium atom to dibenzylic position has a small barrier of 6.3 kcal/mol. These two steps in whole could be considered as 1,4-palladium/hydride shift, which was consistent with experimental observation of deuterium experiments (Fig. 2b). For ring closure of **Int8**, an outersphere displacement of the palladium moiety by the tethered phenolic anion leads to the formation of final product **3**, which suffers a barrier of 12.5 kcal/mol. Our calculation shows the rate-determining step is dediazonation to form the metal carbene species (from **Int2** to **Int5**). Other possible pathways were detailed and discussed in Supplementary Information (See Supplementary Information Fig. S3).

**Synthetic applications.** The catalytic methods for the synthesis of medium-sized lactones were found to be efficient for large scale of reactions (Fig. 4). Thus, when 4 mmol (1 g) of **1c** was treated with the bifunctional diazo precursors under optimal conditions, lactones **3** and **58** were obtained in 74% and 67% of isolated yields, respectively. These lactones could be reduced efficiently by LAH, giving the corresponding diols **83** and **84** in quantitative yields. Intramolecular cyclization of **83** would give dibenzo-oxepine **85**. Diol **84** was converted to suberene **86** by using a consecutive DMP oxidation and TiCl₄-mediated McMurry reaction protocol. To our delight, we found compound **86** has significant utility in preparing miscellaneous medicinally important molecules[49–51]. For example, cyproheptadine **87**, an antihistaminic and anti-serotonergic agent, could be prepared by following reported procedures in three steps[52,53].

## Discussion

In summary, we present a rapid approach to dibenzo-fused seven- and eight-membered lactones. The current palladium-catalyzed medium-sized lactonization is efficient, featuring broad substrate scope, good functional group compatibility, and benefitted by utilization of easily available feedstocks as reactants. From a mechanistic viewpoint, the migratory insertion of palladium carbene is critical, as it afforded the exact palladium(II) intermediate III (Fig. 2d) possessing a fitted geometry to undergo 1,4-palladium shift. This migration process was further supported by DFT calculations. We believe that such a metal/hydride migration could be a general reaction mode to achieve site-selective C–H bond activation. Further studies following this designed principle are ongoing in our laboratories.

## Methods

**General procedure**. An oven-dried Schlenk tube was cooled to room temperature and filled with argon. To this tube was added Pd(OAc)$_2$ (5 mol%), Xantphos (7.5 mol%), K$_2$CO$_3$ (3.0 equiv.), N-tosylhydrazone 2a (0.4 mmol). After the tube was evacuated and refilled with argon three times, 1a (0.2 mmol) and anhydrous THF (2.0 mL) were added. The mixture was stirred at 80 °C. After the reaction was complete (monitored by thin-layer chromatography), the crude mixture was cooled to room temperature, filtered through a short pad of celite and eluented with EtOAc. The resulting solution was concentrated by rotary evaporation. Then, the residual was purified by column chromatography on silica gel to give the pure product 3.

## Data availability

The authors declare that the main data supporting the findings of this study, including experimental procedures and compound characterization, are available within the article and its Supplementary Information files, or from the corresponding author upon request. X-ray structural data of compound 66 are available free of charge from the Cambridge Crystallographic Data Center under the deposition number CCDC 1893958. These data can be obtained free of charge from The Cambridge Crystallographic Data Center via www.ccdc.cam.ac.uk/data_request/cif.

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

## Acknowledgements

We acknowledge financial support by the NSFC (21573237, 21603227, 21871259, 21901244), NSF of Fujian province (2017J05032), the Hundred-Talent Program, and the Strategic Priority Research Program of the Chinese Academy of Sciences (XDB20000000). Professors Armido Studer from Westfälische Wilhelms-Universität Münster, Nuno Maulide from University of Vienna and Hideki Yorimitsu from Kyoto University are greatly acknowledged for their helpful discussion and comments on this work.

## Author contributions

X.H. conceived and directed the project; Y.Y. and X.H. designed the experiments; Y.Y., P.C., and L.Z. performed the experiments; J.S. and C.L. performed the theoretical studies; X.H., Y.Y., and P.C. analyzed all the results and prepared the manuscript.

## Competing interests

The authors declare no competing interests.
