## [Peer Review File · Nature Communications]

Reviewers' comments:

Reviewer #1 (Remarks to the Author):

This manuscript by Huang and co-workers describes the synthesis of 7- and 8-membered dibenzofused lactones through the palladium-catalyzed coupling of ortho-bromo (or -OTf) benzaldehydes and o-hydroxy N-tosylhydrazones. The mechanism of this original reaction involves carbene insertion, 1,4-Pd shift and intramolecular trapping of the resulting acylpalladium intermediate by the ortho-phenol group. The reaction proved efficient on an impressive number of examples (over 80, even though many of them are quite similar), both reaction partners can be synthesized from readily available salicylaldehydes, and the lactone products are valuable scaffolds for medicinal chemistry. On the downside, the manuscript is not very well written and should be carefully edited before publication. In conclusion, this work features a novel and interesting transformation and I recommend acceptance with the minor revision noted below.

1. p. 2: the sentence 'the migration of palladium was proceeded in a bimolecular fashion' is neither grammatically nor semantically correct. The palladium migration is still unimolecular (intramolecular), but occurs after a bimolecular step.
2. Fig. 1 and Fig. 2: the broadly accepted mechanism for the C-H activation step, also reported in the authors' previous work (ref #34) is the concerted metalation-deprotonation whereby the coordinated base abstracts the proton and there is no change of oxidation state at Pd. Hence the cyclopalladation mechanism described in these figures, i. e. oxidative addition to palladium(IV) is not correct and should be modified. To avoid complicated drawings the palladacycles could be modified as Pd•XH instead of H-Pd-X (in the current work X is carbonate).
3. The reference to a 'docking' process for the carbene insertion step throughout the manuscript is unnecessarily confusing. This step has little to do with ligand-receptor interactions in biological contexts. Please remove this and any other related terms (ref #40 is completely unrelated and should be also removed).
4. Table 3:
 - R groups should be used similar to Table 2 to lighten this Table and make it more readable;
 - in which cases was dppb employed instead of Xantphos? Please add a note.
5. 'paracetamol' is misspelled.
6. p. 10: 'isotopic experiments' should be replaced with 'isotopic labelling experiments'.
7. p. 11 and Fig. 2: alternative to attacking palladium to give IV, the phenol could directly attack the C=O to give the lactone with concomitant reductive elimination.
8. p. 13: the term 'immense utility' is overstated and should be moderated to 'significant utility'.

Reviewer #2 (Remarks to the Author):

In this submitted manuscript, the authors disclosed an easy approach for the synthesis of a number of dibenzofused seven-membered lactones by using 1,4-Palladium migration as the key step, where the lactones were constructed by the cross coupling of o-bromoarylaldehydes and various o-hydroxybenzyl N-tosylhydrazones through the formed palladium-carbene enabled acylation in moderate to good yields. Obviously, this protocol provided an efficient way to synthesize the seven-member lactones as mentioned in the context. Whereas, for preparation of the same targeted lactones the Baeyer-Villiger oxidation when unsymmetric cyclic ketones were employed gave the low region selectivity and required the multi-step manipulation. However, as the palladium carbene caused acylation through diazo compound involved C-H bond activation has previously reported by the authors, in which the diazo compounds were employed in Pd-catalyzed cross-coupling reaction leading to lactones. Therefore, the activation of C(sp²)-H bond of an arylaldehyde moiety is known (Angew Chem Int Ed. 2018, 57, 319). In addition, an early work by Larock and co-workers reported the palladium-catalyzed annulation of o-halobenzaldehydes with internal alkynes as the electrophiles, based on the extensively studied migratory insertion of the alkynes occurred with C-H bond activation (Larok et al, J. Org. Chem. 1993, 58, 4579). As regarding to the title, it seems that use of the medium-sized lactones is a quite wide expression, therefor it would be better to referring the products 3 appeared in this manuscript to seven-membered rings. In the meantime, the authors might report that are these any by-product by this protocol, more or less, existing in the reaction products shown in the Table 2 or 3.

I can not see this manuscript reaching the high innovation standard in this journal.

Reviewer #3 (Remarks to the Author):

The present manuscript by Huang and coworkers describes the synthesis of medium-sized lactones by palladium carbene migratory insertion enabled 1,4-palladium shift strategy. It seems that this work was an updated one based on their recent published one (Angew. Chem. Int. Ed. 2018, 57, 319).

Palladium-catalyzed carbene insertions between aryl halides or triflates and N-tosylhydrazones reactions have previously been reported (Chem. Rev. 2017, 117, 13810; Acc. Chem. Res. 2013, 46, 236; OBC, 2016, 3809), as 1,4-palladium shift also has been described in many works (Org. Lett. 2010, 12, 480).

Although this method features broad substrate scope, good functional group compatibility, it does meet the criteria of urgency and novelty required for publication in nature communications.

This reviewer feels that the authors really need to further elaborate this work to publish in the future. Please review the following comments:

1. The authors overstated the title by using 'unprecedented' to present the work.
2. Figure 1 'inert C-H bond functionalization' can't include aldehyde (-CHO)
3. In mechanism section, the colour of the structure should be consistent (Fig.2). What is the driving force in 1,4-Pd shift? Is the intermediate III' Pd (IV) stable? It could be isolated by adding some ligands. Which pathway is more favorable between Pd (IV) and Pd (II)? In order to deep understand the mechanism, this reviewer suggests the authors to isolate intermediate III'.

Reviewer #1:

This manuscript by Huang and co-workers describes the synthesis of 7- and 8-membered dibenzo-fused lactones through the palladium-catalyzed coupling of ortho-bromo (or -OTf) benzaldehydes and o-hydroxy N-tosylhydrazones. The mechanism of this original reaction involves carbene insertion, 1,4-Pd shift and intramolecular trapping of the resulting acylpalladium intermediate by the ortho-phenol group. The reaction proved efficient on an impressive number of examples (over 80, even though many of them are quite similar), both reaction partners can be synthesized from readily available salicylaldehydes, and the lactone products are valuable scaffolds for medicinal chemistry. On the downside, the manuscript is not very well written and should be carefully edited before publication. In conclusion, this work features a novel and interesting transformation and I recommend acceptance with the minor revision noted below.

1. p. 2: the sentence ‘the migration of palladium was proceeded in a bimolecular fashion’ is neither grammatically nor semantically correct. The palladium migration is still unimolecular (intramolecular), but occurs after a bimolecular step.

Response: We have modified the corresponding sentence.

2. Fig. 1 and Fig. 2: the broadly accepted mechanism for the C-H activation step, also reported in the authors’ previous work (ref #34) is the concerted metalation-deprotonation whereby the coordinated base abstracts the proton and there is no change of oxidation state at Pd. Hence the cyclopalladation mechanism described in these figures, i. e. oxidative addition to palladium(IV) is not correct and should be modified. To avoid complicated drawings the palladacycles could be modified as Pd•XH instead of H–Pd–X (in the current work X is carbonate).

7. p. 11 and Fig. 2: alternative to attacking palladium to give IV, the phenol could directly attack the C=O to give the lactone with concomitant reductive elimination.

Response: According to our deuterium labeling experiments, the mechanism of current work is different from our previous work. As describe in the manuscript, we didn’t observe any deuterium labeling of **3** when the reaction was carried out in presence of 10 equiv. of D₂O, which might suggest the involvement of an intermediate akin to Pd•XH is less likely. The deuterium atom labeled in **[D]-1c** was totally transferred to the methylene motif in **[D]-3**. Combined these results with previous studies from prof. Larock (ref. 49 in revised manuscript) and prof. Martin (ref. 21), we thought the reaction pathway depicted in Fig. 2d might be more reasonable.

As suggested by this reviewer, an alternative reaction pathway, involving the addition of phenoxyl anion to the aldehyde moiety, could not be ruled out at this moment.

We have tried very hard on stoichiometric experiments to get more information on the mechanism, but we failed to isolate the palladium intermediates related to III or III’. Further experiments to unveil a clearer picture of current palladium-catalysed lactonization would be another objective of our future studies.

3. The reference to a ‘docking’ process for the carbene insertion step throughout the manuscript is unnecessarily confusing. This step has little to do with ligand-receptor interactions in biological contexts. Please remove this and any other related terms (ref #40 is completely unrelated and should be also removed).

Response: We thank this reviewer's suggestion. This part has been changed.

4. Table 3:

- *R groups should be used similar to Table 2 to lighten this Table and make it more readable;*
- *in which cases was dppb employed instead of Xantphos? Please add a note.*

Response: In table 3, we want to display the formal dimerization/cross-dimerization of salicylaldehyde derivatives. Therefore, all structures of the products were depicted.

For the preparation of 8-membered lactones, dppb works better than Xantphos. We added a note accordingly in table 3.

5. 'paracetamol' is misspelled.

6. p. 10: 'isotopic experiments' should be replaced with 'isotopic labelling experiments'.

8. p. 13: the term 'immense utility' is overstated and should be moderated to 'significant utility'.

Response: These have been revised.

Reviewer #2:

In this submitted manuscript, the authors disclosed an easy approach for the synthesis of a number of dibenzofused seven-membered lactones by using 1,4-Palladium migration as the key step, where the lactones were constructed by the cross coupling of o-bromoarylaldehydes and various o-hydroxybenzyl N-tosylhydrazones through the formed palladium-carbene enabled acylation in moderate to good yields. Obviously, this protocol provided an efficient way to synthesize the seven-member lactones as mentioned in the context. Whereas, for preparation of the same targeted lactones the Baeyer-Villiger oxidation when unsymmetric cyclic ketones were employed gave the low region selectivity and required the multi-step manipulation. However, as the palladium carbene caused acylation through diazo compound involved C-H bond activation has previously reported by the authors, in which the diazo compounds were employed in Pd-catalyzed cross-coupling reaction leading to lactones. Therefore, the activation of C(sp²)-H bond of an arylaldehyde moiety is known (Angew Chem Int Ed. 2018, 57, 319). In addition, an early work by Larock and co-workers reported the palladium-catalyzed annulation of o-halobenzaldehydes with internal alkynes as the electrophiles, based on the extensively studied migratory insertion of the alkynes occurred with C-H bond activation (Larok et al, J. Org. Chem. 1993, 58, 4579).

As regarding to the title, it seems that use of the medium-sized lactones is a quite wide expression, therefor it would be better to referring the products 3 appeared in this manuscript to seven-membered rings.

Response: We respect this reviewer's opinion. The current work depicted here is focused on the reactions for modular preparation of seven- and eight-membered lactones.

In the meantime, the authors might report that are these any by-product by this protocol, more or less, existing in the reaction products shown in the Table 2 or 3.

Response: The main by-products derived from N-tosylhydrazones themselves. That's why in all cases we have excess amount of N-tosylhydrazones (2.0 equiv) to the reaction vessel.

I can not see this manuscript reaching the high innovation standard in this journal.

Reviewer #3:

The present manuscript by Huang and coworkers describes the synthesis of medium-sized lactones by palladium carbene migratory insertion enabled 1,4-palladium shift strategy. It seems that this work was an updated one based on their recent published one (Angew. Chem. Int. Ed. 2018, 57, 319). Palladium-catalyzed carbene insertions between aryl halides or triflates and N-tosylhydrazones reactions have previously been reported (Chem. Rev. 2017, 117, 13810; Acc. Chem. Res. 2013, 46, 236; OBC, 2016, 3809), as 1,4-palladium shift also has been described in many works (Org. Lett. 2010, 12, 480).

Although this method features broad substrate scope, good functional group compatibility, it does meet the criteria of urgency and novelty required for publication in nature communications.

This reviewer feels that the authors really need to further elaborate this work to publish in the future. Please review the following comments:

1. The authors overstated the title by using 'unprecedented' to present the work.

Response: We thank this reviewer's suggestion. This has been revised.

2. Figure 1 'inert C-H bond functionalization' can't include aldehyde (-CHO)

Response: As suggested, this was revised.

3. In mechanism section, the colour of the structure should be consistent (Fig.2). What is the driving force in 1,4-Pd shift? Is the intermediate III' Pd (IV) stable? It could be isolated by adding some ligands. Which pathway is more favorable between Pd (IV) and Pd (II)? In order to deep understand the mechanism, this reviewer suggests the authors to isolate intermediate III'.

Response: The color of the structure in Fig. 2. has been modified.

Considering the exact mechanism, there are several questions need to be addressed by future studies. We are not sure at this moment what is the driving force for 1,4-Pd shift. But according to previous studies from prof. Larock, the formation of thermodynamically stable five-membered palladacycle might be a reason to trigger this kind of palladium migration. The suggestion to isolate intermediate III' could be a solid evidence for the existence of Pd(IV) intermediate. However, up-till-now, the experiments we carried out on this purpose were unsuccessful.

Reviewers' comments:

Reviewer #1 (Remarks to the Author):

This revised manuscript addresses most of my points and concerns. I still have a couple of points which should be addressed for the final version:

1. The term 'docking' is still present on p. 11.
2. Intermediate IV in Fig. 2 is not consistent with the DFT calculations (Fig. 3) which indicate that the phenoxide attacks the carbonyl group and not Pd. Please modify Fig. 2 accordingly.
3. p. 15: replace 'confirmed' with 'supported' in the sentence 'This migration process was further confirmed by DFT calculations'.
4. To answer the authors' comments, I do not think that the deuterium labeling experiments in Fig. 2a-b rule out a CMD process for the 1,4-Pd shift. If the deprotonation/reprotonation steps are faster than intermolecular H-D exchange, then the same outcome will be observed. Please moderate your conclusions.

Reviewer #3 (Remarks to the Author):

Medium-sized lactones widely exist in many natural products and biological activity compounds. Nowadays, the efficient synthesis of these targets received great attentions from academic and industrial chemists. In this manuscript, Huang and coworker disclosed a metal carbene migratory insertion enabled 1,4-palladium shift method to synthesize the medium-sized lactones from two readily available benzaldehyde derivatives. In addition, Huang's method is amenable to late-stage modification of approved drugs and other complex molecules. Moreover, they proposed a reasonable mechanism based on control experiments and DFT calculation. Overall, this work is well written, the science appears to be thorough, and will be of broad interest to the readership of Nature. Common.

Reviewer #4 (Remarks to the Author):

This manuscript addressed the Pd-catalyzed reactions, and experimental part described some new reactions and reaction products, however, the reaction mechanism explored with DFT is only reported with one possibility. In fact, the similar mechanisms for Pd-catalyzed reactions are well reported in the literatures. Therefore, I cannot suggest it publishable on Nature Communication. The main concerns are:

- (1) In Figure 3 (from Ligand exchange to migratory insertion), it should have other possibilities, in my opinion, 2a' should attack int2 to open one of Pd-P first.
- (2) The calculated free-energy barrier is too low, which is not in agreement with experimental condition (353K)

Reviewer #1:

This revised manuscript addresses most of my points and concerns. I still have a couple of points which should be addressed for the final version:

1. The term 'docking' is still present on p. 11.

Response: We have modified the corresponding sentence.

2. Intermediate IV in Fig. 2 is not consistent with the DFT calculations (Fig. 3) which indicate that the phenoxide attacks the carbonyl group and not Pd. Please modify Fig. 2 accordingly.

Response: We have modified the corresponding structure of the intermediate.

3. p. 15: replace 'confirmed' with 'supported' in the sentence 'This migration process was further confirmed by DFT calculations'.

Response: We have modified the corresponding sentence. We thank this reviewer's kind suggestion.

4. To answer the authors' comments, I do not think that the deuterium labeling experiments in Fig. 2a-b rule out a CMD process for the 1,4-Pd shift. If the deprotonation/reprotonation steps are faster than intermolecular H-D exchange, then the same outcome will be observed. Please moderate your conclusions.

Response: We think this reviewer's suggestion is very reasonable, thus we have modified sentence accordingly (see page 12, first paragraph, the sentence that highlighted in yellow color).

Reviewer #3:

Medium-sized lactones widely exist in many natural products and biological activity compounds. Nowadays, the efficient synthesis of these targets received great attentions from academic and industrial chemists. In this manuscript, Huang and coworker disclosed a metal carbene migratory insertion enabled 1,4-palladium shift method to synthesize the medium-sized lactones from two readily available benzaldehyde derivatives. In addition, Huang's method is amenable to late-stage modification of approved drugs and other complex molecules. Moreover, they proposed a reasonable mechanism based on control experiments and DFT calculation. Overall, this work is well written, the science appears to be thorough, and will be of broad interest to the readership of Nature. Common.

Response: We thank this reviewer's comments on our manuscript.

Reviewer #4:

This manuscript addressed the Pd-catalyzed reactions, and experimental part described some new reactions and reaction products, however, the reaction mechanism explored with DFT is only reported with one possibility. In fact, the similar mechanisms for Pd-catalyzed reactions are well reported in the literatures. Therefore, I cannot suggest it publishable on Nature Communication.

Response: We respect this reviewer's comments. As shown in the manuscript, however, our mechanistic studies (especially, the deuterium experiments), clearly support that our work is mechanistically distinct from any other palladium-catalyzed reactions in the literature.

The main concerns are:

(1) In Figure 3(from Ligand exchange to migratory insertion), it should have other possibilities, in

my opinion, 2a' should attack int2 to open one of Pd-P first.

Response: Following this reviewer's suggestion, the possibility of the pathway through attack of **2a'** to **Int2** was further explored. However, all attempts to locate an intermediate with one Pd-P opened with **2a'** failed, presumably because the steric hindrance of ligands prevents the binding of **2a'** to the metal center.

In order to compare with the favorable pathway already shown in the manuscript, we further explored another two possible reaction pathways. As can be seen from the following figure, the direct dissociation of Br⁻ anion from **Int2** to **Int3'** is energetically unfavorable as this process requires the energy of 48.9 kcal/mol. Another process involves the dissociation of one arm of the bidentated ligand to form **Int3** without counter anion exchange, followed by the reaction of **Int3** with **2a'** leading to dediazonation. Such a process is energetically unfavorable as it needs to overcome a large barrier of 38.2 kcal/mol relative to **Int2**. All in all, these two pathways are not favored compared to the one shown in Figure 3 in the manuscript.

(2) The calculated free-energy barrier is too low, which is not in agreement with experimental condition(353K)

Response: We have corrected the free energy at T=353K which is in agreement with our experimental condition. The potential energy surfaces in the manuscript were updated using these new data. The results show that the calculated energy trends do not change and thus the conclusions are not affected. The corrected barrier of rate-determining step is 27.2 kcal/mol, which gives a rate constant of 0.0001 s⁻¹ at 353K according to Eyring equation. This means that the reaction can take place at 353K slowly.

REVIEWERS' COMMENTS:

Reviewer #1 (Remarks to the Author):

The authors addressed my final remarks and hence I am happy to recommend acceptance of this manuscript in the current state.

Reviewer #4 (Remarks to the Author):

No further concerns.